# Prevalence and predictors of depression, anxiety and stress among frontline healthcare workers at COVID-19 isolation sites in Gaborone, Botswana

**Keatlaretse Siamisang** [1,2]\*, **Dineo Kebadiretse** [1,2◉], **Lynn Tuisiree Tjirare** [1,2◉], **Charles Muyela** [1,2◉], **Kebayaone Gare** [1,2◉], **Tiny Masupe** [1]

**1** Department of family Medicine & Public Health, University of Botswana, Gaborone, Botswana, **2** Ministry of Health and Wellness, Gaborone, Botswana

◉ These authors contributed equally to this work.
\* drksiamisang@gmail.com

## Abstract

### Introduction

Coronavirus disease 2019 (COVID-19) has been associated with mental health outcomes and healthcare workers (HCWs) are at the highest risk. The aim of this study was to determine the prevalence and predictors of depression, anxiety and stress, among frontline HCWs at COVID-19 isolation and treatment sites in Gaborone, Botswana.

### Methods

This was a cross-sectional study using self-administered questionnaires at the six (6) isolation facilities. The 42-item Depression, Anxiety and Stress Scale (DASS-42) was used to assess for the outcomes. The proportions are presented with 95% confidence intervals (95% CI). Logistic regression analysis identified predictors of the outcomes. A p value of <0.05 was considered significant.

### Results

A total of 447 participants with a median age of 30 years responded. Depression, anxiety and stress were detected in 94 (21.0% (95% CI 17.3–25.1%)), 126 (28.2% (CI 24.1–32.6%)) and 71 (15.9% (12.6–19.6%)) of the participants respectively. Depression was associated with smoking (AOR 2.39 (95% CI 1.23–4.67)), working at the largest COVID-19 isolation centre, Sir Ketumile Masire Teaching Hospital (SKMTH) (AOR 0.25 (95% CI 0.15–0.43)) and experience of stigma (AOR 1.68 (95% CI 1.01–2.81)). Tertiary education (AOR 1.82 (95% CI 1.07–3.07)), SKMTH (AOR 0.49 (95% CI 0.31–0.77)), household members with chronic lung or heart disease (AOR 2.05 (95% CI 1.20–3.50)) and losing relatives or friends to COVID-19 (AOR 1.72 (95% CI 1.10–2.70)) were predictors of anxiety. Finally, predictors of stress were smoking (AOR 3.20 (95% CI 1.42–7.39)), household members with

**Data Availability Statement:** All relevant data are within the paper and its supporting information file.

**Funding:** The author(s) received no specific funding for this work.

**Competing interests:** The authors have declared that no competing interest exist

chronic heart or lung disease (AOR 2.44 (95% CI 1.27–4.69)), losing relatives or friends to COVID-19 (AOR 1.90 (1.05–3.43)) and working at SKMTH (AOR 0.24 (0.12–0.49)).

## Conclusion

Depression, anxiety and stress are common among frontline HCWs working in the COVID-19 isolation sites in Gaborone. There is an urgent need to address the mental health outcomes associated with COVID-19 including addressing the risk factors identified in this study.

## Background

COVID-19 has affected millions of people across the globe and is duly considered a global health threat [1]. The World Health Organization (WHO) declared the illness a Public Health Emergency of International Concern (PHEIC) on 30th January 2020 and a pandemic on 11th March 2020 [2]. Botswana has not been spared by the COVID-19 pandemic [3]. Several measures were implemented to control the spread of COVID-19 in the country. These measures included appointment of a national COVID-19 task team and restricting movement into the country at points of entry [4]. Contact tracing was implemented with quarantine of close contacts. Confirmed cases were isolated. Sir Ketumile Masire Teaching Hospital (SKMTH) was chosen as the national COVID-19 isolation centre at the beginning of the epidemic. SKMTH is the teaching hospital for the University of Botswana faculty of medicine, situated in the main campus of the university in Gaborone. Prior to the COVID-19 pandemic, it had not been opened for patient care. Almost all of the country's first confirmed cases of COVID-19 were admitted and managed at the hospital. However, as the epidemic progressed many other isolation facilities were utilized including Princess Marina Hospital (PMH), a referral hospital in Gaborone. The other facilities were used by the Greater Gaborone District Health Management Team (DHMT) to isolate relatively stable COVID-19 patients. These patients could be referred to SKMTH or PMH as required.

COVID-19 has significantly affected the mental health of populations across the globe [5]. Not surprisingly, providers caring for patients with COVID-19 are among those at greatest risk of psychological distress [6]. The psychological effects related to the current pandemic are driven by many factors, including uncertainty about the duration of the crisis, lack of proven therapies and potential shortages of healthcare resources such as personal protective equipment (PPE). Furthermore, psychological distress may arise from providing direct care to patients with COVID-19 or knowing someone who has contracted or died of the disease [6]. HCWs are also distressed by the effects of social distancing, and the risk of transmission of the illness to their families [7].

Numerous studies have shown that healthcare professionals are exposed to psychological stressors. Depression, anxiety and stress are particularly common in the context of COVID-19 [2]. In a meta-analysis that had a combined total of 33,062 participants, Pappa et al reported that frontline HCWs during COVID 19 pandemic had a pooled anxiety prevalence of 23.2% and pooled depression prevalence of 22.8% [8]. This suggests that there is a significant burden of psychological effects among HCWs, as a result of novel pandemics, that requires effective and strategically targeted interventions. Stress can cause significant reduction in productivity and reduced performance. This can lead to adverse patient outcomes. Anxiety can lead to loss of confidence and depression which induces more anxiety resulting in a vicious cycle [5]. This

underlines the need to investigate the levels and risk factors of these mental health outcomes. While the psychological impact of COVID-19 has been investigated in several countries, data from Botswana is lacking. The aim of this study was to determine the levels and predictors of stress, anxiety and depression among COVID-19 frontline HCWs in Gaborone, Botswana.

## Methods

### Study sites

The data was collected from the six (6) COVID-19 isolation and treatment centers across Gaborone, the capital City of Botswana. In addition to SKMTH and PMH, the other isolation facilities were Block 8 clinic, Tlotlo hotel, University of Botswana Hotel and Ave Maria Conference Centre.

### Study design

This was a cross sectional study of clinical and non-clinical frontline healthcare workers including but not limited to doctors, nurses, hospital healthcare assistants and cleaners at the COVID-19 isolation and treatment sites in Gaborone, Botswana.

### Selection of subjects

All consenting frontline healthcare workers at the isolation sites were enrolled in the study.

### Sample size calculation

The formula for minimum sample size in prevalence cross sectional study is n = z*z P (1-P)/ d*d where n is the sample size, z is the z statistic, P is the expected prevalence and d is the margin of error. The conventional z statistic of 1.96 for a 95% confidence interval was used. Based on previous studies, the prevalence was expected to fall between 10% and 90% [5,6,9]. The precision was set at 5% or 0.05. A P of 0.5 was chosen to achieve the largest sample size for the specified margin of error. The calculated sample size was 385 participants.

### Data collection

Designated and trained questionnaire administrators collected data from the study sites. Structured self-administered questionnaires were used. Data was collected by trained research assistants in the participants' work environment from 14 July to 24 September 2021. The collected data was shared daily with the investigators. This was to allow daily review of the data and to ensure good data quality. Strict COVID-19 infection prevention and control procedures including wearing of masks, social distancing and use of sanitizers were followed throughout data collection. A self- administered questionnaire was used. The questionnaire included the demographic and medical history data as well as the validated Depression, Anxiety and Stress scale (DASS). DASS is a 42-item questionnaire incorporating 3 subscales of 14 items each [10]. It uses a 4-point Likert scale to rate the extent to which the participants have experienced each state over the past week. The severity of each state is measured from 0 (did not apply to me at all) to 3 (applied to me very much or most of the time). The three dimensions are scored by summing the scores of the relevant items. Based on the total scores, participants were classified as normal, mild, moderate, severe, and extremely severe in each of the domains of depression, anxiety and depression. The cut-off values for these classifications are provided for in the DASS scale [2].

To our knowledge, the DASS scale has not been previously validated in Botswana. However the tool has been widely used and validated in Sub-Saharan Africa. According to a systemic review by Olashore et al, the tool has been used in multiple countries in this region including

Ghana and Ethiopia [11].The tool was pretested on individuals at a district hospital COVID-19 isolation facility. This was used to assess the flow and understanding of the questions as well as assess the internal consistency (reliability) of the data collection tool. The Cronbach alpha coefficient was found to be 0.95, 0.82 and 0.88 for depression, anxiety and stress respectively. The data from this pilot was not included in the final analysis.

### Data analysis

The data was entered into Microsoft Excel. After data cleaning and preparation, IBM Statistical Package for the Social Sciences (SPSS) version 26 was used for data analysis. Categorical data was summarized with frequencies and percentages while numeric data was summarized medians and interquartile ranges. Participants were categorized into normal, mild, moderate, severe and extremely severe according to the cut-off in the respective subscales. The depression subscale is made up of questions 3,5,10,13,16,17,21,24,26,31,34,37,38 and 42. This scale was divided into normal (0–9), mild depression (10–13), moderate depression (14–20), severe depression (21–27) and extremely severe depression (>28). The anxiety subscale is made of questions 2,4,7,9,15,19,20,23,25,28,30,36,40 and 41. The scale was divided into normal (0–7), mild anxiety (8–9), moderate anxiety (10–14), severe anxiety (15–19) and extremely severe anxiety (>20). The rest of the questions make up the stress subscale. The scale was divided into normal (0–14), mild stress (15–18), moderate stress (19–25), Severe stress (26–33) and extremely severe stress (>34).

Bivariate and multiple logistic regression was used to determine risk factors for and protective factors against stress, anxiety and depression (binary variables). A backward conditional method was used to select the most parsimonious model. A p value of <0.05 was considered significant.

### Ethical considerations

The study was approved by the University of Botswana office of research and development (ORD), The Botswana ministry of health and wellness, SKMTH ethics committee, PMH ethics committee and the Greater Gaborone District Health Management Team. Participation in this study was voluntary and written informed consent was obtained from all participants. Privacy and confidentiality were maintained throughout the data collection and management. Only authorized study personnel had access to the data. Participants who were classified as moderate to extremely severe in all of the domains were advised of available resources where they can seek assessment and care.

## Results

### Participants' characteristics

The baseline characteristics of the study participants are displayed in Table 1. A total of, 447 frontline HCWs in Gaborone participated in the study. The median age was 30 years with 271 (60.6%) of participants being female. SKMTH accounted for nearly half of participants at 222 (49.7%) while PMH accounted for 165 (36.9%) of the participants. The rest of the participants were from Block 8 clinic (8.7%), Tlotlo Hotel (2.5%), University of Botswana hotel (1.1%) and Ave Maria Conference centre (1.15). Of the 447 healthcare workers, 156 (34.9%) were nurses, 32 (7.2%) were doctors and 95 (21.3%) were cleaners.

### Participants' medical and social circumstances

The participants' medical history and social circumstances are presented in Table 2. There were 66 (14.8%) HCWs with chronic medical conditions while 51 (11.4%) reported to be on

**Table 1. Characteristics of participants (n = 447).**

| Variable | Number (%) |
|---|---|
| **Sex** | |
| Female | 271 (60.6) |
| Male | 176 (39.4) |
| **Age, median (IQR) (years)** | 30 (25–36) |
| **Age, range (years)** | 20–67 |
| **Marital status** | |
| Single | 360 (80.5) |
| Married | 79 (17.7) |
| Engaged | 1 (0.2) |
| Divorced | 7 (1.6) |
| **Health Facilities** | |
| Sir Ketumile Masire Teaching Hospital | 222 (49.7) |
| Princess Marina Hospital | 165 (36.9) |
| Block 8 clinic | 39 (8.7) |
| Tlotlo Hotel | 11 (2.5) |
| University of Botswana Hotel | 5 (1.1) |
| Ave Maria Conference centre | 5 (1.1) |
| **Job Cadre** | |
| Doctor | 32 (7.2) |
| Nurse | 156 (34.9) |
| Cleaner | 95 (21.3) |
| Healthcare assistant | 85 (19.0) |
| Laundry worker | 20 (4.5) |
| Porter | 19 (4.3) |
| Security | 24 (5.4) |
| Other | 16 (3.6) |
| **Education Level** | |
| No formal Education | 3 (0.7) |
| Primary | 11 (2.5) |
| Secondary | 103 (23.0) |
| Tertiary | 330 (73.8) |
| **Religion** | |
| Christian | 419 (93.7) |
| Muslim | 9 (2.0) |
| African Traditional Religion | 1 (0.2) |
| None | 18 (4.0) |

*IQR Interquartile range.

long-term medications. Smoking and alcohol use were reported by 59 (13.2%) and 143 (32.0%) participants respectively. Participants were asked about having high-risk individuals in their families or households. Household members with chronic heart or lung disease were reported by 77 (17.2%) while 39 (8.7%) reported having household members with cancer. There were 153 (34.2%) participants who reported losing a close friend or relative to COVID-19. A total of 128 (28.6%) of the study participants reported stigma and discrimination due to the nature of their job.

**Table 2. Medical history and participants' social circumstances (n = 447).**

| Variable | Number (%) |
|---|---|
| Chronic Medical Conditions | 66 (14.8) |
| Long term Medications | 51 (11.4) |
| History of psychiatric illness | 18 (4.0) |
| Experience of work-related stigma and discrimination | 128 (28.6) |
| History of depression | 54 (12.1) |
| History of asthma | 40 (8.9) |
| History of heart disease | 17 (3.8) |
| History of Cancer | 24 (5.4) |
| Smoking history | 59 (13.2) |
| Alcohol history | 143 (32.0) |
| Isolation/quarantine of family members | 236 (52.8) |
| Household members with Chronic lung or heart disease | 77 (17.2) |
| Household members with cancer | 39 (8.7%) |
| Household members under 5 years of age | 179 (40.0) |
| Close relatives/friends who died from COVID-19 | 153 (34.2) |

## Prevalence and severity of depression, anxiety and stress among participants

Depression, anxiety and stress were detected in 94 (21.0%), 126 (28.2%) and 71 (15.9%) of the participants respectively (Table 3). Mild depression was detected in 38 (8.5%) while moderate depression affected 24 (5.4%) of the participants. Severe and extremely severe depression affected 17 (3.8%) and 15 (3.4%) participants respectively. There were 32 (7.2%) participants with mild anxiety while 39 (8.7%) had moderate anxiety. Severe and extremely severe anxiety were reported by 27 (6.0%) and 28 (6.3%) respectively. Mild stress was reported by 26 (5.8%) of participants while moderate stress was reported by 21 (4.7%) of participants. There were 18 (4.0%) participants with severe stress and 6 (1.3%) with extremely severe stress.

## Severity of depression, anxiety and stress by gender and isolation site

After stratifying by age, 141 (80.1%) males did not have depression compared to 212 (78.2%) females. Mild depression was present in 9.1% of males compared to 8.1% of females. Moderate depression was seen in 5.1% of males compared to 5.5% of females. Severe and extremely severe depression was seen in 2.8% of males. On the other hand 4.4% and 3.7% of female

**Table 3. Levels of depression, anxiety and stress among frontline healthcare workers in Gaborone, Botswana (n = 447).**

|  | Depression subscale | | Anxiety subscale | | Stress subscale | |
|---|---|---|---|---|---|---|
|  | n | % (95% CI) | n | % (95% CI) | n | % (95% CI) |
| Normal | 353 | 79.0 (74.9–82.7) | 321 | 71.8 (67.4–75.9) | 376 | 84.1 (80.4–87.4) |
| Abnormal | 94 | 21.0 (17.3–25.1) | 126 | 28.2 (24.1–32.6) | 71 | 15.9 (12.6–19.6) |
| Mild | 38 | 8.5 (6.1–11.5) | 32 | 7.2 (4.9–10.0) | 26 | 5.8 (3.8–8.4) |
| Moderate | 24 | 5.4 (3.5–7.9) | 39 | 8.7 (6.3–11.7) | 21 | 4.7 (2.9–7.1) |
| Severe | 17 | 3.8 (2.2–6.0) | 27 | 6.0 (4.0–8.7) | 18 | 4.0 (2.4–6.3) |
| Extremely Severe | 15 | 3.4 (1.9–5.5) | 28 | 6.3 (4.2–8.9) | 6 | 1.3 (0.5–2.9) |

* DASS: Depression, Anxiety and Stress Scale.

**Table 4. Depression, anxiety and stress levels according to gender and study site.**

| | Gender | | Study site (facility) | | |
|---|---|---|---|---|---|
| | Male | Female | SKMTH | PMH | Block 8 Clinic |
| **Depression level** | | | | | |
| No depression | 141 (80.1) | 212 (78.2) | 200 (90.1) | 110 (66.7) | 28 (71.8) |
| Mild depression | 16 (9.1) | 22 (8.1) | 11 (5.0) | 24 (14.5) | 3 (7.7) |
| Moderate depression | 9 (5.1) | 15 (5.5) | 6 (2.7) | 10 (6.1) | 4 (10.3) |
| Severe depression | 5 (2.8) | 12 (4.4) | 4 (1.8) | 11 (6.7) | 2 (5.1) |
| Extremely severe depression | 5 (2.8) | 10 (3.7) | 1 (0.5) | 10 (6.1) | 2 (5.1) |
| **Anxiety level** | | | | | |
| No anxiety | 128 (72.7) | 193 (71.2) | 178 (80.2) | 107 (64.8) | 24 (61.5) |
| Mild anxiety | 14 (18.0) | 18 (6.6) | 18 (8.1) | 8 (4.8) | 4 (10.3) |
| Moderate anxiety | 21 (11.9) | 18 (6.6) | 14 (6.3) | 17 (10.3) | 5 (12.8) |
| Severe anxiety | 8 (4.5) | 19 (7.0) | 9 (4.1) | 14 (8.5) | 2 (5.1) |
| Extremely severe anxiety | 5 (2.8) | 23 (8.5) | 3 (1.4) | 19 (11.5) | 4 (10.3) |
| **Stress level** | | | | | |
| No stress | 152 (86.4) | 224 (82.7) | 208 (93.7) | 121 (73.3) | 32 (82.1) |
| Mild stress | 9 (5.1) | 17 (6.3) | 7 (3.2) | 14 (8.5) | 4 (10.3) |
| Moderate stress | 8 (4.5) | 13 (4.8) | 5 (2.3) | 13 (7.9) | 1 (2.6) |
| Severe stress | 4 (2.3) | 14 (5.2) | 2 (18.2) | 12 (7.3) | 2 (5.1) |
| Extremely severe stress | 3 (1.7) | 3 (1.1) | 0 (0) | 5 (3.0) | 0 (0) |

participants had severe and extremely severe depression respectively. Similarly, 72.7 of males had no anxiety compared to 71.2% of female participants. There were more male participants (86.4%) with no stress than female participants (82.7%). Participants working at SKMTH had the lowest rates of depression, anxiety and stress. Out of the 222 participants from this facility, 200 (90.1%) had no depression, 178 (80.2%) had no anxiety and 208 (93.7%) had no stress. In contrast, PMH participants had high rates of depression, anxiety and stress. Of the 165 participants from PMH, 110 (66.7%) had no depression, 107 (64.8%) had no anxiety and 121 (73.3%) had no stress (Table 4).

### Factors associated with depression, anxiety and stress among the frontline HCWs

Table 5 shows the predictors of depression in the bivariate and multivariate regression models. The statistically significant predictors of depression in the multivariate model were history of smoking (AOR 2.39, p = 0.010), working at SKMTH (AOR 0.25, p< 0.001) and experience of stigma or discrimination due to line of work (AOR 1.68, p = 0.049).

Independent predictors of anxiety were tertiary education (AOR 1.82, p = 0.026), working at SKMTH (AOR 0.49, p = 0.002), having household or family members with chronic lung

**Table 5. Predictors of Depression among frontline Healthcare workers in Gaborone.**

| Variable | Crude Odds Ratio | p value | Adjusted odds ratio | p value |
|---|---|---|---|---|
| History of smoking* | 1.98 (1.09–3.62) | 0.026 | 2.39 (1.23–4.67) | 0.010 |
| Having household members or family with chronic lung disease | 2.26 (1.32–3.89) | 0.003 | 1.78 (0.99–3.20) | 0.052 |
| Working at Sir Ketumile Masire Teaching Hospital* | 0.23 (0.14–0.39) | <0.001 | 0.25 (0.15–0.43) | <0.001 |
| Experience of stigma/discrimination* | 1.79 (1.11–2.89) | 0.018 | 1.68 (1.01–2.81) | 0.049 |
| Close relatives or friends died of COVID-19 | 2.20 (1.39–3.50) | 0.001 | 1.59 (0.96–2.61) | 0.071 |

**Table 6. Predictors of anxiety among frontline healthcare workers in Gaborone.**

| Variable | Crude odds ratio | p value | Adjusted odds ratio | p value |
|---|---|---|---|---|
| Working as a doctor | 0.56 (0.23–1.41) | 0.223 | 0.46 (0.18–1.19) | 0.111 |
| Tertiary education* | 1.65 (1.00–2.72) | 0.050 | 1.82 (1.07–3.07) | 0.026 |
| Working at Sir Ketumile Masire Teaching Hospital* | 0.43 (0.28–0.66) | <0.001 | 0.49 (0.31–0.77) | 0.002 |
| Having household members or family with chronic lung disease* | 2.39 (1.44–3.97) | 0.001 | 2.05 (1.20–3.50) | 0.008 |
| Relatives or friends died of COVID-19* | 2.13 (1.39–3.25) | <0.001 | 1.72 (1.10–2.70) | 0.018 |

disease (AOR 2.05, p = 0.008) and losing close relatives and friends due to COVID-19 (AOR 1.72, p = 0.018). See Table 6 below.

Finally, independent predictors of stress were history of smoking (AOR 3.20, p = 0.005), having household members with chronic heart or lung disease (AOR 2.44, p = 0.007), losing close relatives and friends to COVID-19 (AOR 1.90, p = 0.034) and working at SKMTH (AOR 0.24, <0.001). See Table 7 below.

## Discussion

COVID-19 has had a significant impact on mental health particularly of frontline HCWs. We set out to determine the prevalence and predictors of stress, depression and anxiety as well as their severity levels among frontline HCWs at COVID-19 isolation and treatment sites in Gaborone, Botswana. The study population included doctors, nurses, cleaners and others who were involved in the care of patients with COVID-19. As expected, the majority of the participants were from the 2 referral centres for COVID-19 patients in Gaborone, Botswana. About a third of the participants were nurses. This was expected as this profession has the highest density compared to other cadres. According to Nkomazana et al, in 2014, the density of nurses per 10,000 in Botswana was 41.3 compared to 4.3 for doctors [12]. It is therefore expected that there are more nurses than other cadres in the Botswana COVID-19 frontline. This is significant as there is evidence that nurses may be at increased risk of mental health outcomes than other cadres [13].

The observed burden of the mental health outcomes was generally lower than what has been reported elsewhere. The prevalence of depression, anxiety and stress were 50.4%, 44.6% and 71.5% respectively in a Chinese cross sectional study during the early stages of the COVID-19 pandemic [14]. A systematic review mainly involving Asian countries reported a pooled prevalence of 23.2% for anxiety disorders and 22.8% for depression [8]. The current study prevalence of depression and anxiety are within the range of the pooled prevalence. In contrast, a systematic review involving nine Northern, Eastern Western and Southern African countries reported a prevalence of anxiety disorders ranging from 9.5% to 73.3% while depression prevalence ranged from 12.5% to 71.9% [11]. Our findings fall within the ranges of the

**Table 7. Predictors of stress among frontline healthcare workers in Gaborone.**

| Variable | Crude Odds Ratio | p value | Adjusted odds ratio | p value |
|---|---|---|---|---|
| Male sex | 0.76 (0.43–1.34) | 0.340 | 0.53 (0.27–1.05) | 0.082 |
| Working as a cleaner | 0.36 (0.15–0.86) | 0.021 | 0.43 (0.16–1.17) | 0.098 |
| History of Smoking* | 2.20 (1.13–4.31) | 0.021 | 3.20 (1.42–7.39) | 0.005 |
| Having household members with chronic lung or heart disease* | 3.01 (1.66–5.47) | <0.001 | 2.44 (1.27–4.69) | 0.007 |
| Relatives or friends died of COVID-19* | 2.71 (1.57–4.68) | <0.001 | 1.90 (1.05–3.43) | 0.034 |
| Working at Sir Ketumile Masire Teaching Hospital* | 0.20 (0.10–0.39) | <0.001 | 0.24 (0.12–0.49) | <0.001 |

African systematic review. A cross sectional study in Ethiopia showed similar results to our study with a prevalence of 20.2%, 21.9% and 15.5% for depression, anxiety and stress respectively [15]. Conversely, another study in Ethiopia had a slightly higher prevalence of stress of 31.4% while the prevalence of depression and anxiety were 25.8% and 36% respectively [16]. The incongruity of the results may be due to the use of different scales and different cut-off scores to measure depression, anxiety and stress. Furthermore socioeconomic and cultural environment, workload, variation in the availability of personal protective equipment and the difference in mental preparedness related to previous epidemics may also contribute to the observed difference. Additionally, our study was done relatively later in the epidemic. The HCWs may have adjusted or may have received care for mental health symptoms. There is evidence that mental health outcomes are associated with the stage of the epidemic or point in the epidemic curve [8]. As the COVID-19 epidemic evolves in Botswana, follow up studies would be useful in determining the variability of these mental health outcomes.

A subgroup analysis based on gender showed that females were more likely to experience depression, anxiety and stress. This is consistent with previous studies [17–19]. While the risk of adverse outcomes may be higher for males, females have been shown to have a more pronounced psychological response to the COVID-19 pandemic [8]. More studies are needed to explore the association between gender and psychological outcomes in our setting. This is important, as most of the frontline HCWs in our study were females. In the current study, HCWs at SKMTH had lower prevalence of depression, anxiety and stress compared to other isolation facilities. SKMTH was the primary treatment site for COVID -19 and hence the hospital may have been well resourced and the HCWs may have been mentally prepared to deal with the epidemic as compared to other sites. Indeed HCWs at SKMTH have had access to wellness sessions at the facility. They also had better access to PPE and were provided with accommodation away from their loved ones. Moreover the hospital enjoyed technical and other support from the University of Botswana faculty of medicine.

Several predictors of depression, anxiety and stress were identified in this study. Smoking was significantly associated with both depression and stress. The odds of depression were more than twice among participants with a history of smoking. Smokers were also more than three (3) times likely to have stress. Other studies have found an association between smoking and mental health outcomes. Smoking was significantly associated with depression and anxiety among Bangladesh HCWs who smoked during COVID-19 [19]. Similarly, a study conducted in Greece in the pre-COVID era also showed that smoking among HCW is associated with depression and anxiety [20]. More recently, in a study of mental health of a community in New Zealand, Gasteiger and colleagues determined that history of smoking was a significant predictor of anxiety during the first 10 weeks of COVID-19 [21]. Our study was conducted when HCWs had become adept at managing COVID-19. This may explain why smoking was not a significant predictor of anxiety. It is worth noting that the literature on the association between smoking and mental health outcomes is inconsistent. This was demonstrated by a systematic review by Fluharty et al who found inconsistent results in the association between smoking and depression and anxiety [22]. Another predictor of depression in HCWs is experiencing stigma or discrimination due to working with highly infectious patients. This is consistent with the findings in Bangladesh where discrimination in the workplace and social challenges due to the HCWs involvement in COVID-19 patients' care were associated with anxiety and depression [19].

Having a household or family with chronic lung or heart disease was significantly associated with depression, anxiety and stress. Nearly one-fifth of the participants in our study had family or household members with these co-morbidities. The perceived risk to these vulnerable and high-risk individuals raises much uncertainty about the hazard HCWs pose to them

particularly when they live in close proximity to them. Our study demonstrates that the mental health outcome are not just about the HCWs' concern for their personal risk but also risk to their loved ones. HCWs working at SKTMH were significantly less likely to have depression, anxiety or stress than the others. This may be due to better access to personal protective equipment as well as access to wellness and counseling sessions. Likewise, SKTMH staff has been managing COVID-19 patients for longer. They also do not have to manage general patients. On the other hand, the other facilities have to balance COVID-19 with management of other conditions.

Having completed tertiary education was significantly associated with anxiety. This is consistent with findings from an Indian study investigating mental health outcomes among frontline HCWs [23]. The more educated HCWs have more knowledge and understanding of the disease. One would expect that knowledge would allay uncertainty and fears about a disease. On the other hand, more knowledge about risk may exacerbate anxiety. Although many studies have shown that being a female HCW is associated with anxiety, this was not the case in the current study. Wilson et al reported two-fold increased odds of anxiety, depression and stress among Indian female HCWs compared to males [24]. Similarly, studies among HCWs in China, Bangladesh, Turkey and Egypt also demonstrated significant association between female gender and anxiety [19,25,26]. The lack of significant association in the current study may be due to the timing of the data collection. Since the data was collected relatively late in the stage of the COVID-19 pandemic, many female HCWs may have already received care.

## Limitations

This was a cross-sectional study with self-reporting of symptoms in the past week. This could have missed symptoms experienced earlier and underestimated the mental health outcomes in this population. Furthermore, the data was collected more than a year after the first cases were reported in Botswana. At this stage, the participants had likely adjusted to the disease. The results could have been different if the data had been collected early in the pandemic. This limits comparison to early studies.

While the sampling was exhaustive, voluntary participation could potentially introduce selection bias if the people who were eager to participate were systematically different from the ones who declined. Possible confounding was managed by multiple regression analysis. However only known confounders can be corrected for.

## Conclusion

Depression, anxiety and stress are prevalent among frontline HCWs based in the COVID-19 isolation and treatment sites in Botswana. Multiple risk factors were identified including history of smoking, working at SKMTH, experience of stigma and discrimination and having close relatives or friends who died of COVID-19, attainment of tertiary education and having household members with chronic lung disease. There is an urgent need to address the mental health issues associated with COVID-19 including addressing the risk factors identified in this study. Addressing the mental health of HCWs should be part of the COVID-19 response in every healthcare facility.

## Supporting information

**S1 File.**
(XLSX)

## Acknowledgments

We would like to thank the management and staff at all the COVID-19 isolation sites for their support. We would also like to thank the University of Botswana public health medicine unit and the Ministry of Health and Wellness. We would like to thank the following people for their role in data collection: Lindiwe Montle, Lebogang Maphane, Melanie Iris Hagen, Thata Losike, Olebile Sekabodile, Wendy Brooks and Ngwao Ngwako.

## Author Contributions

**Conceptualization:** Keatlaretse Siamisang, Tiny Masupe.

**Data curation:** Keatlaretse Siamisang.

**Formal analysis:** Keatlaretse Siamisang.

**Investigation:** Keatlaretse Siamisang, Dineo Kebadiretse, Lynn Tuisiree Tjirare, Charles Muyela, Kebayaone Gare, Tiny Masupe.

**Methodology:** Keatlaretse Siamisang, Lynn Tuisiree Tjirare, Charles Muyela, Kebayaone Gare, Tiny Masupe.

**Project administration:** Keatlaretse Siamisang, Dineo Kebadiretse, Charles Muyela, Kebayaone Gare, Tiny Masupe.

**Supervision:** Keatlaretse Siamisang, Dineo Kebadiretse, Lynn Tuisiree Tjirare, Charles Muyela, Kebayaone Gare, Tiny Masupe.

**Visualization:** Keatlaretse Siamisang.

**Writing – original draft:** Keatlaretse Siamisang.

**Writing – review & editing:** Dineo Kebadiretse, Lynn Tuisiree Tjirare, Charles Muyela, Kebayaone Gare, Tiny Masupe.

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
