## [Decision Letter · Decision Letter 0]

17 Jun 2022

PONE-D-22-14292Prevalence and predictors of depression, anxiety and stress among frontline healthcare workers at COVID-19 isolation sites in Gaborone, BotswanaPLOS ONE

Dear Dr. SIAMISANG,

Thank you for submitting your manuscript to PLOS ONE. After careful consideration, we feel that it has merit but does not fully meet PLOS ONE’s publication criteria as it currently stands. Therefore, we invite you to submit a revised version of the manuscript that addresses the points raised during the review process.

We look forward to receiving your revised manuscript.

Kind regards,

Orvalho Augusto, MD, MPH

Academic Editor

PLOS ONE

Journal Requirements:

Additional Editor Comments:

This report adds numbers to the topic so far, suspected or subjectively discussed. Mental health issues are quite common among healthcare workers in general and especially the frontline healthcare workers. Despite assessing one year after the first case was notified, here the authors did conduct a cross-sectional assessment of depression, anxiety and stress among frontline health workers in health the facilities designated to be the COVID-19 treatment and isolation centres in Gaberone/Botswana. They used the DASS survey that has been used elsewhere, but not as much in the context of Gaberone (I do hope the authors provide more on this).

In general, this is a very well-written manuscript. Here are a few issues:

1. Abstract in the results for both the prevalence and the OR please add their confidence intervals.

2. Line 111/112 - what is this exhaustive sampling.

3. Lines 120/121 - There is a little language issue. Usually, when we do not know where the real proportion would be we choose the proportion of 0.5 because this is where the proportion gets the largest variance i.e the largest sample size for a certain margin of error.

4. Line 140: it seems it should be depression, anxiety and stress. Correct, please.

5. Lines 142 to 147: was this tool tested in Gaberone/Botswana previously?

6. Somewhere in the data analysis section, we miss a clear indication of how you did dichotomize the scales. In fact, we do not know how you did classify each person into each of the 6 levels (like in table 3), to begin with.

7. Table 3: I would suggest adding confidence intervals for the prevalence at least for the abnormal row.

8. Put below the table the abbreviations and the name of the test used

9. Sadly, again, we use inferior methods to analyse ordinal variables (depression, anxiety and stress levels). For example, the Chi-square applied in table 4 just checks whether there is one cell particularly different from the expected or not. Such a test discards the fact that the depression level has an order, and we are not informed what is the direction of an association if it exists. This is just a lament. I am not requesting the authors to change the analysis.

Reviewers' comments:

Reviewer's Responses to Questions

**Comments to the Author**

1. Is the manuscript technically sound, and do the data support the conclusions?

Reviewer #1: Yes

2. Has the statistical analysis been performed appropriately and rigorously? 

Reviewer #1: Yes

3. Have the authors made all data underlying the findings in their manuscript fully available?

Reviewer #1: Yes

4. Is the manuscript presented in an intelligible fashion and written in standard English?

Reviewer #1: Yes

5. Review Comments to the Author

Reviewer #1: Line 103:The second sentence in the paragraph is not important as in the next paragraph the participants have been defined

Line 111: I would avoid using the phrase the sampling was exhaustive but rather describing what was done to get an exhaustive sampling

line 114 sample size calculation: Not clear why the authors decided to use a range for expected prevalence. Though I imagined the authors have obtained a good sample size, I would have advised using a prevalence from a previous study that was done in a geographical location closest to the study site.

Table 4: I am sure the importance of the p value in the table 4, this is purely a descriptive statistics of the participants.

6. PLOS authors have the option to publish the peer review history of their article (what does this mean?). If published, this will include your full peer review and any attached files.

Reviewer #1: **Yes: **Stewart Ngasa

---

## [Author Response · Author response to Decision Letter 0]

20 Jul 2022

Response to Reviewers

1. Abstract in the results for both the prevalence and the OR please add their confidence intervals.

Confidence intervals have been added for both the prevalence and odds ratios.

2. Line 111/112 - what is this exhaustive sampling.

The phrase “exhaustive sampling” has been removed and is replaced with a statement of how the sampling was done which is that all consenting eligible participants were enrolled.

3. Lines 120/121 - There is a little language issue. Usually, when we do not know where the real proportion would be we choose the proportion of 0.5 because this is where the proportion gets the largest variance i.e the largest sample size for a certain margin of error.

This has been corrected. It now reads “Based on previous studies, the prevalence was expected to fall between 10% and 90%. The precision was set at 5% or 0.05. A P of 0.5 was chosen to achieve the largest sample size for the specified margin of error. The calculated sample size was 385 participants”

4. Line 140: it seems it should be depression, anxiety and stress. Correct, please.

This has been corrected.

5. Lines 142 to 147: was this tool tested in Gaberone/Botswana previously?

To the best of our knowledge, the tool has not been tested in Botswana. It has however been validated in other Sub Saharan countries. We tested the tool through a pilot. We have described this pilot in the paper.

We have cited a systematic review that reports use of this tool in other Sub-Saharan countries.

6. Somewhere in the data analysis section, we miss a clear indication of how you did dichotomize the scales. In fact, we do not know how you did classify each person into each of the 6 levels (like in table 3), to begin with.

We have now described in detail how the different levels of the 3 domains were calculated in the analysis section. The statement reads, “Participants were categorized into normal, mild, moderate, severe and extremely severe according to the cut-off in the respective subscales. The depression subscale is made up of questions 3,5,10,13,16,17,21,24,26,31,34,37,38 and 42. This scale was divided into normal (0-9), mild depression (10-13), moderate depression (14-20), severe depression (21-27) and extremely severe depression (>28). The anxiety subscale is made of questions 2,4,7,9,15,19,20,23,25,28,30,36,40 and 41. The scale was divided into normal (0-7), mild anxiety (8-9), moderate anxiety (10-14), severe anxiety (15-19) and extremely severe anxiety (>20). The rest of the questions make up the stress subscale. The scale was divided into normal (0-14), mild stress (15-18), moderate stress (19-25), Severe stress (26-33) and extremely severe stress (>34).

7. Table 3: I would suggest adding confidence intervals for the prevalence at least for the abnormal row.

The confidence intervals have been added for all the proportions in table 3.

8. Put below the table the abbreviations and the name of the test used

This has been done

9. Sadly, again, we use inferior methods to analyse ordinal variables (depression, anxiety and stress levels). For example, the Chi-square applied in table 4 just checks whether there is one cell particularly different from the expected or not. Such a test discards the fact that the depression level has an order, and we are not informed what is the direction of an association if it exists. This is just a lament. I am not requesting the authors to change the analysis.

We have removed the p values as per recommendation by the reviewer. (see below)

 If there are no restrictions, please upload the minimal anonymized data set necessary to replicate your study findings as either Supporting Information files or to a stable, public repository and provide us with the relevant URLs, DOIs, or accession numbers.

We have uploaded the dataset as supporting information.

Reviewer #1: 

1.Line 103:The second sentence in the paragraph is not important as in the next paragraph the participants have been defined

The sentence has been removed

2.Line 111: I would avoid using the phrase the sampling was exhaustive but rather describing what was done to get an exhaustive sampling

The phrase was removed

3.line 114 sample size calculation: Not clear why the authors decided to use a range for expected prevalence. Though I imagined the authors have obtained a good sample size, I would have advised using a prevalence from a previous study that was done in a geographical location closest to the study site.

We used a P of 0.5 or 50% to get the maximum sample size for the specified margin of error. The calculated sample size is adequate and higher than it would be with a known prevalence. 

Table 4: I am sure the importance of the p value in the table 4, this is purely a descriptive statistics of the participants.

We have removed the p value in table 4.

---

## [Editor Report · Decision Letter 1]

2 Aug 2022

Prevalence and predictors of depression, anxiety and stress among frontline healthcare workers at COVID-19 isolation sites in Gaborone, Botswana

PONE-D-22-14292R1

Dear Dr. SIAMISANG,

We’re pleased to inform you that your manuscript has been judged scientifically suitable for publication and will be formally accepted for publication once it meets all outstanding technical requirements.

Kind regards,

Orvalho Augusto, MD, MPH

Academic Editor

PLOS ONE
---

## [Editor Report · Acceptance letter]

5 Aug 2022

PONE-D-22-14292R1 

Prevalence and predictors of depression, anxiety and stress among frontline healthcare workers at COVID-19 isolation sites in Gaborone, Botswana 

Dear Dr. SIAMISANG:

I'm pleased to inform you that your manuscript has been deemed suitable for publication in PLOS ONE. Congratulations! Your manuscript is now with our production department. 

Kind regards, 

on behalf of

Dr. Orvalho Augusto 

Academic Editor

PLOS ONE